# PopAlign: Population-Level Alignment for Fair Text-to-Image Generation

**Shufan Li, Harkanwar Singh, Aditya Grover**
{jacklishufan,harkanwarsingh,adityag}@cs.ucla.edu
University of California, Los Angeles

## Abstract

Text-to-image (T2I) models achieve high-fidelity generation through extensive training on large datasets. However, these models may unintentionally pick up undesirable biases of their training data, such as over-representation of particular identities in gender or ethnicity neutral prompts. Existing alignment methods such as Reinforcement Learning from Human Feedback (RLHF) and Direct Preference Optimization (DPO) fail to address this problem effectively because they operate on pairwise preferences consisting of individual *samples*, while the aforementioned biases can only be measured at a *population* level. For example, a single sample for the prompt "doctor" could be male or female, but a model generating predominantly male doctors even with repeated sampling reflects a gender bias. To address this limitation, we introduce PopAlign, a novel approach for population-level preference optimization, while standard optimization would prefer entire sets of samples over others. Using human evaluation and standard image quality and bias metrics, we show that PopAlign significantly mitigates the bias of pretrained T2I models while largely preserving the generation quality.

## 1 Introduction

Modern image generative models, such as the Stable Diffusion [18, 27] and DALLE [17, 16, 11] model series, are trained on large datasets of billions of images scraped from the internet. As a result, these models tend to strongly inherit various kinds of biases from their dataset. For example, in Figure 1a, we can see that SDXL tends to generate predominantly male images for the prompt "doctor".

In this work, we study a specific category of biases that are defined at a *population* level. That is, a single sample from a generative model is insufficient to assess whether the model exhibits a specific population bias. Prominent examples include biases of text-to-image generative models with respect to gender or ethnicity neutral prompts. For example, a single generated image sample for the prompt "doctor" could be male or female, but a model generating predominantly male doctors even with repeated sampling reflects a gender bias.

This contrasts with much of the recent AI safety and alignment work for large language models [4, 35], where the harmfulness in generations can be ascertained at the level of individual samples. For example, given the prompt "What is the gender of doctors?", even individual generated text responses should ideally not show a bias towards a specific gender.

Given any implicit population preference (e.g., equalizing image generations across genders for a gender-neutral prompt), there are two key challenges in aligning large-scale text-to-image generative models. First, many state-of-the-art models are trained on large-scale, possibly non-public datasets, making it prohibitively expensive for intermediate developers to retrain them for population alignment.

Safe Generative AI Workshop
38th Conference on Neural Information Processing Systems (NeurIPS 2024).

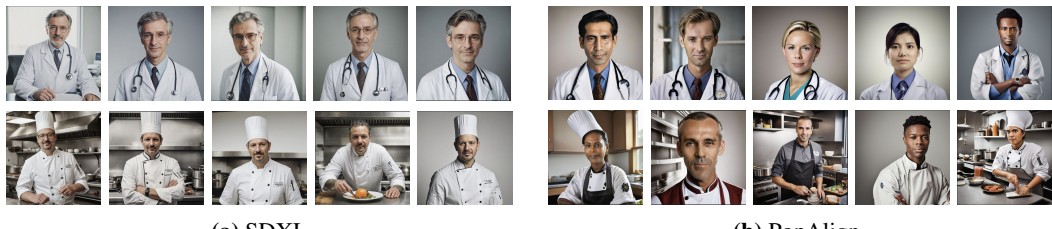

| **(a)** SDXL | **(b)** PopAlign |

**Figure 1:** Illustration of PopAlign, our proposed framework for mitigating the bias of pretrained T2I models using population-level alignment. **Left:** SDXL over-represents a particular identity as it picked up biases of the training data. **Right:** PopAlign mitigates the biases without compromising the quality of generated samples.

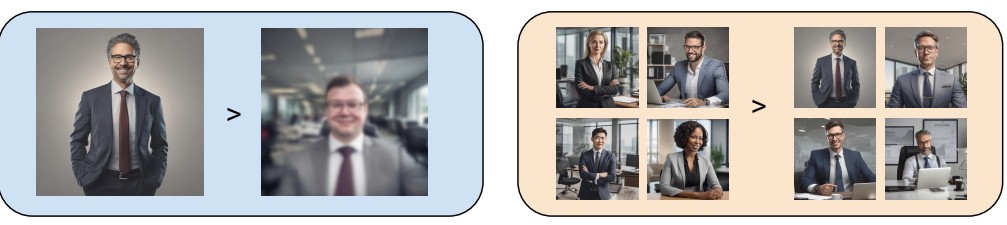

**(a)** Sample-level preferences (RLHF/DPO)  **(b)** Population-level preferences (PopAlign)

**Figure 2:** Difference between PopAlign and existing RLHF/DPO Methods. **Left:** Existing methods such as RLHF/DPO use pairwise preferences of individual samples to improve image quality. **Right** PopAlign uses population-level preferences to achieve better fairness and diversity.

Therefore, an ideal solution would build on existing models, be sample-efficient in acquiring additional supervision, and parameter-efficient for cost-effective alignment. Second, given the diverse range of concepts represented in modern generative models, population alignment on a specific dimension (e.g., gender) should not degrade visual quality for any kind of prompt.

Our primary contribution in this work is to define PopAlign, a preference alignment framework for mitigating population bias for text-to-image generative models. Building on the RLHF framework, we first propose to acquire multi-sample preferences over sets of samples, as proxies for population-level preferences. We reduce it to a corresponding reward-free, population-level DPO objective.

Finally, we derive the PopAlign objective as a stochastic lower bound to this population-level DPO objective such that it permits tractable evaluation and maximization by decomposing multi-sample pairwise preferences into single-sample preferences after sampling from their respective populations.

To evaluate our model's efficacy, we collect population-level preference data through a combination of human labelers and automatic pipelines based on attribute classifiers. Through standard image quality and bias metrics as well as extensive human evaluations, we show that PopAlign significantly mitigates bias in pretrained text-to-image models without notably impacting the quality of generation. Compared with a base SDXL model, PopAlign reduces the gender and race discrepancy metric of the pretrained SDXL by (-0.233), and (-0.408) respectively, while maintaining comparable image quality.

## 2  Method

Consider a pretrained text-to-image model $\pi_\theta$ that is biased w.r.t. one or more population-level traits. Our goal in population-level alignment is to fine-tune PopAlign *without* acquiring any additional real images. To do so, we assume access to a source of preferences (e.g., via humans) over the model's output generations.

## 2.1 Population-Level Preference Acquisition

Typically, alignment data for RLHF/DPO is created by generating multiple samples using the same prompt and asking humans to rank the results. Since the goal of PopAlign is to mitigate the population-level bias, we need to generate two or more *sets* of images for the same prompt. However, naive sampling of sets does not work due to the high degree of bias within current T2I models for identity-neutral prompts. For example, we observe that among 100 images generated from the prompt "doctor", only 6 are female doctors This makes generating a set of near-fair samples nearly impossible using this naive method.

To address this challenge, we use an approximated process where we directly augment a gender-neutral prompt such as "engineering" to a diverse set of identity-specific prompts such as "Asian male engineer" and "female engineer", and use images sampled from these augmented prompts as the *winning set*, and images sampled directly from the gender-neutral prompt as the *losing set*. As a sanity check, for each pair of sets, we use a classifier in combination with a face detector to determine if the sampled images are indeed consistent with the prompts.

## 2.2 Population-Level Alignment from Human Preferences

Given a prompt $c$ and two sets of generated images $X_0, X_1$ where $|X_0| = |X_1| = N$, The Bradley-Terry (BT) model [1] for human preference is $p^*(X_0 \succ X_1|c) = \sigma(r(X_0, c) - r(X_1, c))$, where $r(X, c)$ is a real-valued reward function dependent on the prompt and the set of generated images.

In the RLHF setup [12], $r(X, c)$ is modeled by a neural network $\phi$ trained on a dataset $\mathcal{D}$ with pairs of winning samples and losing samples $(X^w, X^l, c)$ by optimizing the following objective function:

$$\mathcal{L}_r(r_\phi, \mathcal{D})) = -\mathbb{E}_{c, X^w, X^l \sim \mathcal{D}}[\log \sigma(r(X_w, c) - r(X_l, c))]. \tag{1}$$

Once the reward model is trained, we can optimize a generative model $\pi_\theta$ using the PPO objective:

$$\max_{\pi_\theta} \mathbb{E}_{c \sim \mathcal{D}, x_1, ..x_N \sim \pi_\theta(x|c)}[r(\{x_1, ..x_N\}, c)] - \beta \mathbb{D}_{\mathrm{KL}}[\pi_\theta(X|c)||\pi_{\mathrm{ref}}(X|c)] \tag{2}$$

where $X = x_1, ..x_N$ is a population of generated samples and $\pi_{\mathrm{ref}}$ is a reference distribution. Typically, $\pi_{\mathrm{ref}}$ is a pretrained model and $\pi_\theta$ is initialized with $\pi_{\mathrm{ref}}$. Further, using an analogous derivation as DPO [15], we know that the optimal solution of Eq. (2), say $\pi_\theta^*$ satisfies the condition $r^*(X, c) = \beta \log \frac{\pi_\theta^*(X|c)}{\pi_{\mathrm{ref}}(X|c)} + \beta \log Z(c)$, where Z(c) is the partition function. Combining this with Eq. (1), we obtain an equivalent objective:

$$\max_{\pi_\theta} \mathbb{E}_{c, X^w, X^l \sim \mathcal{D}}[\log \sigma(\beta \log \frac{\pi_\theta(X^w|c)}{\pi_{\mathrm{ref}}(X^w|c)} - \beta \log \frac{\pi_\theta(X^l|c)}{\pi_{\mathrm{ref}}(X^l|c)})]. \tag{3}$$

Using this objective, we can directly optimize $\pi_\theta$ without explicitly training a reward model.

## 2.3 Population Level Alignment of Text-to-Image Diffusion Models

In the context of text-to-image diffusion models, the winning and losing population $X^w, X^l$ each consists of $N$ images generated independently through the diffusion process $\{x^{w,i}\}_{i=1,2..N}, \{x^{l,i}\}_{i=1,2..N}$. Hence, we can rewrite Eq. (3) as:

$$\max_{\pi_\theta} \mathbb{E}_{c, X^w, X^l \sim \mathcal{D}}[\log \sigma(\beta \log \frac{\prod_{i=1}^N \pi_\theta(x^{w,i}|c)}{\prod_{i=1}^N \pi_{\mathrm{ref}}(x^{w,i}|c)} - \beta \log \frac{\prod_{i=1}^N \pi_\theta(x^{l,i}|c)}{\prod_{i=1}^N \pi_{\mathrm{ref}}(x^{l,i}|c)})]. \tag{4}$$

Naively using this objective can be computationally expensive, because it requires computing the distribution of all samples in the set at the same time. However, we can further establish a lower bound of this objective by applying Jensen's inequality on the concave function $\log \sigma(x)$:

$$\max_{\pi_\theta} \mathbb{E}_{c, x \sim X, X \sim \mathcal{D}, t \sim \mathrm{Uni}(\{1, 2..T\}), i \sim \mathrm{Uni}(\{1, 2..N\})}[\log \sigma(\gamma_X \beta' \log \frac{\pi_\theta(x_{t-1}|x_t, c)}{\pi_{\mathrm{ref}}(x_{t-1}|x_t, c)} - \gamma_X \beta' \mu)] \tag{5}$$

where $\mathrm{Uni}()$ denotes the uniform distribution, $\gamma_X$ is an indicator with value +1 when $X$ is a winning population and -1 when $X$ is a losing population, $\beta' \propto \beta$ is a constant, $\mu$ is a normalizer, $x_t$ are sampled from a diffusion process. We provide a full proof of the derivation in Appendix G. This formulation allows us to train the model effectively without computing the whole diffusion process at each step. Empirically, we set $\mu = \mathbb{E}[\log \frac{\pi_\theta(x_{t-1}|x_t, c)}{\pi_{\mathrm{ref}}(x_{t-1}|x_t, c)}]$ estimated through batch statistics.

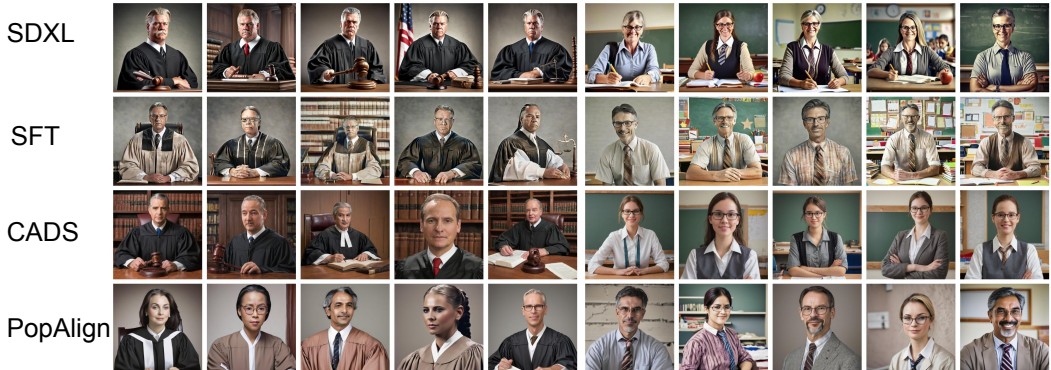

**Figure 3:** Qualitative results on gender-neutral prompts. PopAlign mitigates the bias of the pretrained SDXL in both male-skewed or female-skewed prompts.

## 3 Experiments

We conducted experiments with SDXL [13], a state-of-the-art T2I as the base model. We consider two aspects of biases: gender and race. For fairness, we use the fairness discrepancy metric $f$ proposed by earlier works [2], which measures fairness on sensitive attribute $u$ over individual image samples $x$ as

$$f(p_{\text{ref}}, p_\theta) = \|\mathbb{E}_{p_{\text{ref}}}[p(u|x)] - \mathbb{E}_{p_\theta}[p(u|x)]\|_2 \tag{6}$$

where $p_{\text{ref}}$ is an ideal distribution and $p_\theta$ is the distribution of a generative model. The lower is the discrepancy metric, the better can the model mitigate unfair biases. We use the DeepFace library, which contains various face detection and classification models for this metric [22, 23, 24, 25],

For image quality, we employ a set of standard image quality metrics: CLIP [14], HPS v2 [32], and LAION aesthetics score [20]. We evaluate the performance of our method on a set of 100 gender neural prompts. These prompts are manually written and are different from the training prompts. For each prompt, we generate 100 images, achieving a total sample size of 10,000. We report the discrepancy metric on gender and race, as well as image quality metrics in Table 1, and provide qualitative results in Fig. 3

**Table 1:** Results on gender-neutral and ethnic-neutral prompts.

|  | Discrepancy | | Quality | | |
| --- | --- | --- | --- | --- | --- |
|  | Gender↓ | Race↓ | HPS ↑ | Aesthetic ↑ | CLIP ↑ |
| SDXL | 0.417 | 0.666 | 25.2 | 5.66 | **28.2** |
| SDXL-SFT | 0.307 | 0.471 | 21.6 | 5.72 | 21.3 |
| SDXL-PopAlign | **0.184** | **0.258** | **25.9** | **5.84** | 28.2 |
| SDXL-CADS[19] | 0.334 | 0.641 | 21.5 | 5.83 | 26.3 |
| SDXL-Dynamic-CFGS[19] | 0.307 | 0.552 | 22.5 | 5.76 | 26.4 |
| SDXL-aDFT[26] | 0.251 | 0.307 | 22.0 | 5.68 | 22.4 |
| SDXL-Iti-gen[34] | 0.257 | 0.314 | 25.1 | 5.43 | 27.9 |
| SDXL-Fair-Diffusion[6] | 0.195 | - | 24.7 | 5.77 | 25.0 |
| SDXL-DPO | 0.294 | 0.642 | **34.6** | 5.71 | 31.5 |
| SDXL-DPO-PopAlign | 0.189 | **0.331** | 33.2 | 5.84 | 31.4 |

## 4 Conclusions

In summary, we propose PopAlign, a novel algorithm that mitigates the biases of pretrained text-to-image diffusion models while preserving the quality of the generated images. PopAlign successfully extend the pair-wise preference formulation used by RLHF and DPO to a novel population-level alignment objective, surpassing comparable baselines.

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

# A Related Works

## A.1 Diversity and fairness in image generation

Diversity and fairness are active areas of research in image generation. However, these terminologies often refer to distinct concepts in past works. The word diversity is used to refer broadly to the coverage of concepts in the training distribution. Accordingly, many techniques exist to improve diversity. For example, in current diffusion models, we can tune the guidance [5, 7] as a knob for trading off diversity with image quality. However, these works as well as recent extensions (e.g., [8], [19]) focus on diversity as a generic term, and not diversity of specific attributes that have fairness and equity implications such as race and gender. For examples, for the prompt "doctor", a set of images of white male doctors with varying hairstyles, camera angles, lighting conditions, backgrounds can be considered as more "diverse" than generating a single image of a middle-aged doctor with the same pose and background. While indeed diverse along one axis, this notion does not capture "fair" representation of identities, which is the focus of this work.

Another line of related works focus on "fairness", which measures whether generative samples matches a desired distribution over a specific sets sensitive attributes such as gender and race. We discuss some representative works Early approaches that reweigh the importance of samples in a biased training dataset to improve fairness [2]. FairGen [28] improve the fairness of a pretrained Generative adversarial network (GAN) by shifting its latent distribution using Gaussian mixure models. FairTL [29] improves the fairness of GAN by fine-tuning a discriminator on a small unbiased dataset. Um and Suh [30] employs LC-divergence to improve the fairness of GAN, which better captures the distance between real and generated in small training datasets. Despite their successes, these methods are tested on small datasets such as CelebA [10]. They are not applicable to T2I diffusion models pretrained on large-scale datasets either because of GAN specific designs or requires re-training using the pretrained data. Most recently, FairDiffsuion attempts to mitigate the bias of diffusion model at *deployment* time by maintaining a lookup table of known biases (e.g. "prompt doctor is biased towards male"), and injecting image-editing prompts at inference time. By contrast, our method address the problem at model *release*. It is also more flexible and does require maintaining a lookup table of known biases.

## A.2 Aligning generative models with human preferences

A growing line of recent work considers the *alignment* of the outputs of large language models (LLMs) to improve their safety and helpfulness by directly querying humans (or other AI models) to rank or rate model outputs to create a preference dataset. The most basic approach is reinforcement learning with human preferences (RLHF) [3], which trains a reward model on this preference data and then employs reinforcement learning to maximize the expected rewards. The RL step typically make use of proximal policy optimization (PPO) [21] to prevent the model from diverging too much from the pretrained model. DPO [15] simplified this process by converting the RL objective to a supervised-finetuning-style objective, eliminating the need to first fit a reword model. Recently, various works [31, 33] extended DPO to text-to-image diffusion models. These works mostly focus on improving the quality of generated images, with little emphasis on fairness and safety.

# B Training Details

## B.1 PopAlign

We use ChatGPT to generate 300 identity-neutral prompts involving no specific gender or race, such as "a botanist cataloging plant species in a dense forest" and "a biochemist examining cellular structures, in a high-tech lab". We augment the prompt with gender and race keywords as described in Sec. 2.1. In particular, we consider gender keywords "male" and "female" and race keywords "white","Asian","black","Latino Hispanic","Indian","middle eastern" as specified by the classifier. It should be noted that this list is not an exhaustive representation of all possible identities. However, our method can easily be generalized to incorporate other diversities with appropriate prompts. We generate 100 images for each identity-neutral prompts and 10 images for each identity-specific prompts. Afterwards, we obtain set-level preference data as described in Sec. 2.1. While images

can be generated by either identity-neutral or identity-specific prompts in our pipeline, we use the identity-neutral prompt as the caption label in the training data.

We train our models using 4 Nvidia A5000 GPUs for 750 iterations. We use a per-GPU batch size of 2. We employ AdamW optimizer with a learning rate of 5e-07 for 750 iteration

## B.2 Baselines

We compare our models against other methods that aims to address fairness in T2I generation, namely aDFT[26], Iti-Gen[34] and Fair-Diffusion[6]. We would like to note that comparing with these methods are non-trivial as they use different classifiers (CLIP, Fairface, etc.), different base models (SDv1.0, SDv1.5, etc.), different ways of categorizing races, and different evaluation protocols (aDFT uses 50 prompts, ITI-Gen uses 5 prompts).

To make the comparison fair and relevant, we adopted the following setups: 1.We employ state-of-the-art diffusion model SDXL as the base model 2.We adopt the same classifier as PopAlign, which classifies images into two genders and five races. We only use this classifier to replace explicit classifiers where needed. For methods that use CLIP text encoder and CLIP features as part of the formulation or loss, we keep the CLIP model intact. 3.We evaluate the baselines on (1) a set of 100 identity-neutral prompts specified in Sec 6.3 (2) 600 identity-specific prompts.

Additionally, we made the following additional adjustments to each method so they work properly in our setup.

**Adjusted-DFT[26]:** This method finetunes the text-encoder to mitigate the bias of conditioning signals. Since SDXL comes with two text-encoders OpenCLIP-ViT/G and CLIP-ViT/L, we train both of them jointly.

**Iti-gen[34]:** This method injects extra learnable embedding after the token embedding layer of the text-encoder. We inject embedding for both text-encoders used in SDXL.

**Fair Diffusion[6]:** Fair Diffusion's formulation only works for binary labels, and the authors discovered that non-binary categories tend to "result in fragile behavior." It is nontrivial to fix this issues and extending the method to non-binary categories. Hence, we only incorporate results of gender edits.

## C  Additional Experiments

**Table 2:** Results on identity-specific prompts.

|  | Recall | | | Quality | | |
| --- | --- | --- | --- | --- | --- | --- |
|  | Gender↑ | Race↑ | Overall↑ | HPS↑ | Aesthetic↑ | CLIP↑ |
| SDXL | 100.0 | 99.8 | 99.8 | 36.7 | 6.05 | 33.6 |
| SDXL-SFT | 100.0 | 95.1 | 95.1 | 35.6 | 5.96 | 33.1 |
| SDXL-PopAlign | 99.0 | 98.8 | 98.0 | 36.8 | 6.09 | 33.4 |
| SDXL-Iti-Gen | 85.6 | 35.7 | 31.3 | 29.3 | 5.75 | 30.2 |
| SDXL-FairDiffusion | 75.9 | - | - | 31.5 | 5.78 | 30.3 |
| SDXL-aDFT | 99.8 | 89.8 | 89.5 | 36.1 | 5.84 | 32.9 |
| SDXL-DPO | 99.8 | 99.8 | 99.6 | 38.2 | 6.20 | 33.8 |
| SDXL-DPO-PopAlign | 100.0 | 99.8 | 98.8 | 37.8 | 6.27 | 33.5 |

## C.1  Identity-Specific Prompts

To verify that our model do not over-generalize superficial diversity for identity-specific prompts, we evaluate our method against the pretrained model and SFT baseline on a set of identity-specific prompts. This is crucial because a model that misrepresents a particular identity when explicitly prompted to do so will raise equity and fairness concerns and is not safe to deploy in an end-user product. We create this specific prompts by augmenting the identity neutral prompts in **??** with

**Table 3:** Results on generic prompts from Pick-a-Pick test set. These prompts are not necessarily gender-neutral and ethnic-neutral.PopAlign was able to maintain the image quality on generic prompts.

| Model | PickScore↑ | HPS ↑ | Aesthetic ↑ | CLIP ↑ |
|---|---|---|---|---|
| SDXL | 21.9 | 36.2 | 5.87 | 32.8 |
| SDXL-SFT | 21.3 | 33.9 | 5.76 | 31.6 |
| SDXL-PopAlign | 21.9 | 35.4 | 5.89 | 32.3 |
| SDXL-DPO | 22.3 | 37.2 | 5.89 | 33.4 |
| SDXL-DPO-PopAlign | 22.4 | 37.2 | 5.90 | 33.2 |

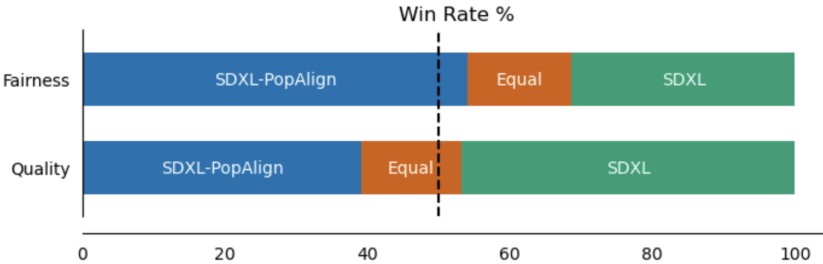

**Figure 4:** Human Evaluation on fairness and quality of the image population

identity keywords such as "female", "Asian". To measure the image-prompt alignment, we report the recall rate of gender and race classifier. Specifically, we classify each of the generated images and check if the classification results match the prompt. We also report image quality metrics including HPS v2, LAION aesthetics and CLIP. We show these results in Table 2.

Almost all methods achieve high scores in recall metrics, suggesting training to mitigate biases on identity-neutral prompts do not adversely affect the generation results of identity-specific prompts. However, SDXL-SFT suffers a slightly larger drop in the overall recall than SDXL-PopAlign. In terms of image quality, we observed a similar pattern as in identity-specific prompts, where PopAlign better preserve than image quality of pretrained models than SFT baselines, as measured in HPS (+1.5), Aesthetic (+0.13) and CLIP (+0.7).

## C.2   Generic Prompts in the Wild

For optimal classification accuracy, we use simple prompts for experiments in **??** and Appendix C.1. However, these prompts are vastly different than complicated prompts typically used by human users. To provide a more comprehensive evaluation of generation quality of PopAlign and pretrained SDXL, we evaluate our models on Pick-a-Pick test set [9] consisting of prompts written by actual humans. These prompts are not necessarily identity-neutral. In fact, some prompts do not include humans at all. In addition to HPS, LAION aesthetics and CLIP metrics, we additionally report the PickScore which is commonly used on this benchmark. We show results in Table 3. These results are consistent with previous experiments. SDXL-PopAlign was able to match the performance of pretrained SDXL, and achieves higher image quality of than SFT baselines.

## C.3   Human Evaluation

We ask the user to judge the fairness of quality of images generated by SDXL and SDXL-PopAlign. We use the images generated using the 100 identity-neutral prompts used in Table 1. The images are grouped into sets of 5 images. We show the results in Fig. 4. Humans generally consider PopAlign a superior model in terms of fairness, and the two models are roughly comparable in terms of image quality.

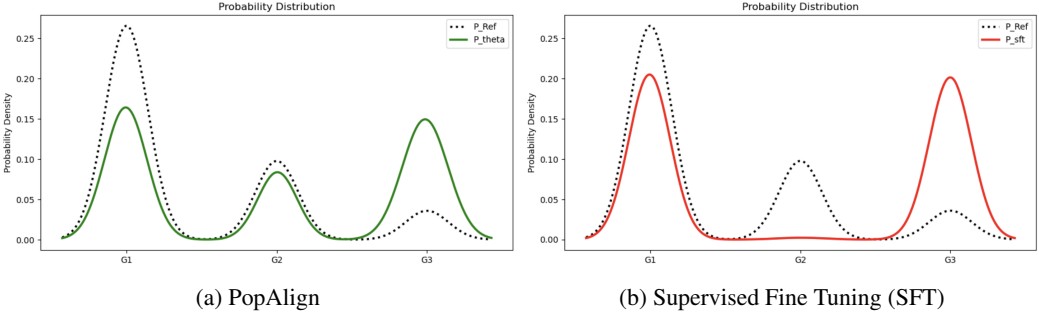

|  (a) PopAlign | (b) Supervised Fine Tuning (SFT) |

**Figure 5:** Effect of PopAlign and SFT on a skewed 1-d distribution **Left:** PopAlign effectively balance the skewed distribution. **Right:** Supervised Fine-Tuning (SFT) result in collapse of the Gaussian not in preference data

## D  Synthetic Evaluation

To verify the behavior of our objective, we also conduct experiments on 1D mixture of Gaussians. In this simple setup, the reference distribution contains three Gaussians G1, G2, and G3, with a high skew between G1 and G3. G1, G3 is analogous to a pair of biased attributes such as "male", "female" where G3 is under-represented. G2 is analogous to an unrelated distribution, such as "trees" or "buildings". We collect 1000 samples to create a population-level preference dataset. The preference dataset do not contain samples from G2, just as our preference data do not contain non-human prompts.

We use Eq. (7) to represent $P_\theta$. We initialize two models with $P_{\text{Ref}}(w_{\text{Ref}} = \text{softmax}(1, 0, -1)$ , $\mu_{\text{Ref}} = (-7.0, 0.0, 7.0)$, $\sigma_{\text{Ref}} = (1.0, 1.0, 1.0))$ . We apply PopAlign (with $\beta$=0.5) and SFT loss to the models respectively and train the model until convergence.

$$P_\theta = \sum_{i=1}^{3} w_i \cdot \mathcal{N}(x; \mu_i, \sigma_i^2) \text{ s.t. } \sum_{i=1}^{3} w_i = 1, \quad \theta = \{w_i, \mu_i, \sigma_i^2 \mid i = 1, 2, 3\} \qquad (7)$$

We show results in Fig. 5 we observe that PopAlign is able to mitigate the biases between G1,G3, while maintaining the distribution of G2. While SFT also balanced on G1,G3, it's support collapses on G2. These results are a simple illustration PopAlign's ability to mitigate the bias while maintaining the generative capability of the model gained from the pretraining data.

## E  Ablation Studies

To validate our design choices, we conducted extensive ablation studies on various hyper-parameters.

### E.1  Classifier-Free Guidance

Classifier free guidance (CFG) is the used to ensure the generated images accurately follow the text prompts. Typically, higher guidance strength leads to sharper images and better image-prompt alignment, at the cost of sample diversity. We show effects of varying CFG on identity-neutral prompts in Fig. 6. For SDXL, higher CFG leads to higher discrepancy, indicating less diversity as expected. However, for SFT and PopAlign, increasing CFG do not significantly compromise the discrepancy because of extra training. Among these two methods, PopAlign consistently exhibits a lower discrepancy. For main experiments, we used a cfg of 6.5.

### E.2  Divergence Penalty

The divergence Penalty $\beta$ is an important hyperparameter as it controls the strength of divergence penalty. We show the results of $\beta = 1000$, $\beta = 3000$ and $\beta = 5000$ in Fig. 7. In general, higher $\beta$ leads to higher image quality as stronger divergence penalty prevents the model from deviating too much from the pretrained checkpoint. This comes with a cost of higher discrepancy. We pick

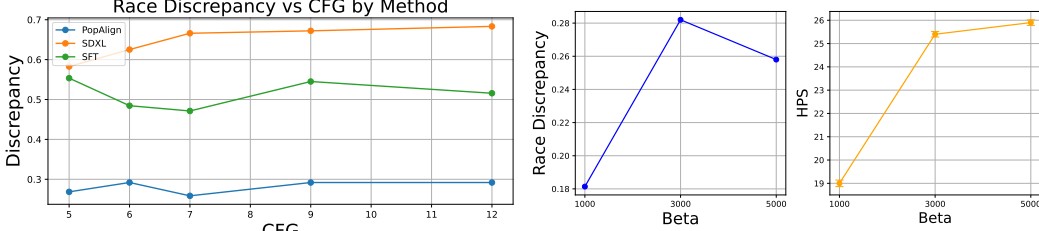

**Figure 6:** Ablation study of varying CFGs

**Figure 7:** Ablation study on divergence penalty $\beta$

$\beta = 5000$ for our experiments, but end-users may choose an alternative based on the relative importance of fairness and image quality.

### E.3  Normalization Factor

Because we remove pair wise preferences in Eq. (5), we need to center the inner term (reward) by $\mu$. Following the conventional practice of RL, we use the expected value of inner term $\mu = \mathbb{E}[\log \frac{\pi_\theta(x_{t-1}|x_t,c)}{\pi_{\text{ref}}(x_{t-1}|x_t,c)}]$, which can be re-written as the weight sum of the expected reward of all positive samples and that of all negative samples $\mu(\alpha) = \alpha\mathbb{E}[\log \frac{\pi_\theta(x_{t-1}^w|x_t^w,c)}{\pi_{\text{ref}}(x_{t-1}^w|x_t^w,c)}] + (1-\alpha)\mathbb{E}[\log \frac{\pi_\theta(x_{t-1}^l|x_t^l,c)}{\pi_{\text{ref}}(x_{t-1}^l|x_t^l,c)}]$ with $\alpha = 0.5$. We also experimented with two alternatives $\alpha = 0.25$ and $\alpha = 0.75$. $\alpha = 0.25$ will move the $\mu$ closer to the side of losing samples, while $\alpha = 0.75$ will move the $\mu$ closer to the side of winning samples. Since the gradient of $\log \sigma$ is symmetric with respect to the origin, and monotonically decreases as it moves away from the origin. $\alpha = 0.25$ will increase the update step of the negative samples because $\mu(0.25)$ is closer to the negative samples, which makes the inner term closer to the origin. Similarly, $\mu(0.25)$ will increase the update step of the positive samples. We show the results in Table 4.

$\alpha = 0.25$ leads to model divergence, as the negative samples have a stronger "pushing force" than the "pulling force" of positive samples in this setup. $\alpha = 0.75$ leads to lower discrepancy and image quality, as it increases the "pulling force" of positive samples, implicitly decreasing the effect of divergence penalty.

**Table 4:** Ablation study of normalization factor.

|  | Discrepancy | | Quality | | |
| --- | --- | --- | --- | --- | --- |
| $\alpha$ | Gender↓ | Race↓ | HPS ↑ | Aesthetic ↑ | CLIP ↑ |
| 0.25 | Diverge | Diverge | -0.5 | 4.25 | 16.3 |
| 0.5 | 0.184 | 0.258 | **25.9** | **5.84** | **28.2** |
| 0.75 | **0.170** | **0.222** | 23.7 | 5.72 | 26.4 |

## F  Human-in-the-Loop Sampling

Our method make uses of a pretrained classifier. However, this representation over simplifies the complex nuances of the real world. In this section, we discuss a human-in-the loop sampling process that may be used to capture these complex aspects of diversity (e.g. non-binary genders, homosexual versus heterosexual couples).

In particular, we consider the setup where human start with a set of initial samples $X^0 = \{x_1^0...x_N^0\}$ where N is the set size. A Markov chain iteratively refines the set of samples $X^t$ to $X^{t+1}$ until it reaches a target distribution. At each step, a random sample $x_i^t$ is replaced with a new sample $x_i^{t+1}$, and we obtain a new set $X^{t+1} = \{x_1^t...x_{i-1}^t, x_i^{t+1}, x_{i+1}^t, x_N^t\}$. We accept the new set if the new set is closer to the target distribution, and back-track otherwise. In the human-in-the-loop version, a human judge will accept the changes that improve the perceived "fairness". For example, in a male-dominate population, the judge will accept changes that replace a male sample with a female sample, and

reject changes in the other direction. The judge will terminate the process when we achieved a fair balance between the male and female samples. Assuming humans and classifiers are reasonably capably of distinguishing the basic properties in question, this human-in-the-loop MCMC should generate a fair distribution with sufficiently large samples as the training data. However, because finding human annotators qualified to fairly evaluate these nuances and curate a large collection of data can be prohibitively expansive, we left this for future works.

## G   Proof of Population Level Alignment Objective

We start with Eq. (4). Following Diffusion-DPO [31], we can substitute $\pi_\theta(x|c)$ with $\sum_{t=1}^{T} \pi_\theta(x_t|x_{t+1}, c)$ and obtain

$$= \mathbb{E}_{c,X^w,X^l \sim \mathcal{D}}[\log \sigma(\beta \log \frac{\prod_{i=1}^{N} \prod_{t=1}^{T} \pi_\theta(x_t^{w,i}|x_{t+1}^{w,i}, c)}{\prod_{i=1}^{N} \prod_{t=1}^{T} \pi_{\text{ref}}(x_t^{w,i}|x_{t+1}^{w,i}, c)} - \tag{8}$$

$$\beta \log \frac{\prod_{i=1}^{N} \prod_{t=1}^{T} \pi_\theta(x_t^{l,i}|x_{t+1}^{l,i}, c)}{\prod_{i=1}^{N} \prod_{t=1}^{T} \pi_{\text{ref}}(x_t^{l,i}|x_{t+1}^{l,i}, c)})] \tag{9}$$

$$= \mathbb{E}_{c,X^w,X^l \sim \mathcal{D}}[\log \sigma(\beta \sum_{i=1}^{N} \sum_{t=1}^{T} \log \pi_\theta(x_t^{w,i}|x_{t+1}^{w,i}, c) - \beta \sum_{i=1}^{N} \sum_{t=1}^{T} \log \pi_{\text{ref}}(x_t^{w,i}|x_{t+1}^{w,i}, c) - \tag{10}$$

$$\beta \sum_{i=1}^{N} \sum_{t=1}^{T} \log \pi_\theta(x_t^{l,i}|x_{t+1}^{l,i}, c) + \beta \sum_{i=1}^{N} \sum_{t=1}^{T} \log \pi_{\text{ref}}(x_t^{l,i}|x_{t+1}^{l,i}, c))] \tag{11}$$

By Jensen's inequality, we have a lower bound

$$\mathbb{E}_{c,X^w,X^l \sim \mathcal{D}}[\log \sigma(\beta \sum_{i=1}^{N} \sum_{t=1}^{T} \log \pi_\theta(x_t^{w,i}|x_{t+1}^{w,i}, c) - \beta \sum_{i=1}^{N} \sum_{t=1}^{T} \log \pi_{\text{ref}}(x_t^{w,i}|x_{t+1}^{w,i}, c) - \tag{12}$$

$$\beta \sum_{i=1}^{N} \sum_{t=1}^{T} \log \pi_\theta(x_t^{l,i}|x_{t+1}^{l,i}, c) + \beta \sum_{i=1}^{N} \sum_{t=1}^{T} \log \pi_{\text{ref}}(x_t^{l,i}|x_{t+1}^{l,i}, c))] \tag{13}$$

$$\geq NT\mathbb{E}_{c,x^w,x^l \sim \mathcal{D}, t \in \text{Unif}(\{1,2..T\}), i \in \text{Unif}(\{1,2..N\})}[\log \sigma(\beta \log \pi_\theta(x_t^{w,i}|x_{t+1}^{w,i}, c) - \tag{14}$$

$$\beta \log \pi_{\text{ref}}(x_t^{w,i}|x_{t+1}^{w,i}, c) - \beta \log \pi_\theta(x_t^{l,i}|x_{t+1}^{l,i}, c) + \beta \log \pi_{\text{ref}}(x_t^{l,i}|x_{t+1}^{l,i}, c))] \tag{15}$$

$$= NT\mathbb{E}_{c,x^w,x^l \sim \mathcal{D}, t \in \text{Unif}(\{1,2..T\}), i \in \text{Unif}(\{1,2..N\})}[\log \sigma(\beta \log \frac{\pi_\theta(x_t^{w,i}|x_{t+1}^{w,i}, c)}{\pi_{\text{ref}}(x_t^{w,i}|x_{t+1}^{w,i}, c)} - \tag{16}$$

$$\beta \log \frac{\pi_\theta(x_t^{l,i}|x_{t+1}^{l,i}, c)}{\pi_{\text{ref}}(x_t^{l,i}|x_{t+1}^{l,i}, c)})] \tag{17}$$

$$= NT\mathbb{E}_{c,x^w,x^l \sim \mathcal{D}, t \in \text{Unif}(\{1,2..T\}), i \in \text{Unif}(\{1,2..N\})}[\log \sigma(\beta \log \frac{\pi_\theta(x_t^{w,i}|x_{t+1}^{w,i}, c)}{\pi_{\text{ref}}(x_t^{w,i}|x_{t+1}^{w,i}, c)} - \tag{18}$$

$$\mu + \mu - \beta \log \frac{\pi_\theta(x_t^{l,i}|x_{t+1}^{l,i}, c)}{\pi_{\text{ref}}(x_t^{l,i}|x_{t+1}^{l,i}, c)})] \tag{19}$$

$$\geq 2NT\mathbb{E}_{c,x \sim X, X \sim \mathcal{D}, t \in \text{Unif}(\{1,2..T\}), i \in \text{Unif}(\{1,2..N\})}[\log \sigma( \tag{20}$$

$$\gamma_X \beta \log \frac{\pi_\theta(x_t|x_{t+1}, c)}{\pi_{\text{ref}}(x_t|x_{t+1}, c)} - \gamma_X \beta \mu)] \tag{21}$$

$$= \mathbb{E}_{c,x \sim X, X \sim \mathcal{D}, t \in \text{Unif}(\{1,2..T\}), i \in \text{Unif}(\{1,2..N\})}[\log \sigma(\gamma_X \beta' \log \frac{\pi_\theta(x_t|x_{t+1}, c)}{\pi_{\text{ref}}(x_t|x_{t+1}, c)} - \gamma_X \beta' \mu)] \tag{22}$$

where $\beta' = 2NT\beta$ and $\mu$ is a normalizing constant to stabilize the optimization.

# H  Details of Human Evaluation

We use the following prompt for human evaluation

---

**Human Evaluation Prompt**

Select the set of images that represents more diversity of identity representation and quality of image.

Look at the two sets of images below generated from a prompt. Each set contains multiple images. Set A is the top 5 images, while Set B is the bottom 5. Select which set you think shows greater diversity in terms of identity representation and quality of image set.

Please consider the variety in elements such as color, subject matter, race, gender, and other visible identity markers when making your selection.

**Which set is more diverse and fair in terms of identity representation:**

- Set A (Top Row) is more diverse and fair
- Set B (Bottom Row) is more diverse and fair
- Both sets are equally diverse and fair

**Which set has better quality images overall:**

- Set A (Top Row) is higher quality
- Set B (Bottom Row) is higher quality
- Both sets are equally good in terms of quality

---

For each pair of sets, we collect responses from three individual human evaluators to mitigate potential noises in human preference. We do not expose human evaluators for any NSFW content. We employ Amazon MTurk for this job. The works are paid with a prorated hourly minimum wage. We follow all guidelines and rules of respective institutions.

# I  Additional Qualitative Results

We provide additional qualitative results in Fig. 8. The samples are generated using the prompt "engineer" and "artist". Compared with the baselines, PopAlign offers a diverse representation of identities while maintaining a comparable image quality with the pretrained SDXL checkpoint.

# J  Additional Experiments

On 1D mixture of Gaussian, the divergence penalty $\beta$ is an important hyperparameter as it controls the strength of divergence penalty. We show the results of $\beta = 0.1$, $\beta = 0.5$ and $\beta = 0.9$ in Fig. 9. In general, higher $\beta$ leads to stronger divergence penalty prevents the model from deviating too much from the pretrained checkpoint. This comes with a cost of higher discrepancy. We pick $\beta = 0.5$ for our 1-d experiments.

We also experimented with $\alpha = 0.25$, $\alpha = 0.5$, $\alpha = 0.75$, $\alpha = 0.25$ will move the $\mu$ closer to the side of losing samples, while $\alpha = 0.75$ will move the $\mu$ closer to the side of winning samples. Since the gradient of $\log \sigma$ is symmetric with respect to the origin, and monotonically decreases as it moves away from the origin. $\alpha = 0.25$ will increase the update step of the negative samples because $\mu(0.25)$ is closer to the negative samples, which makes the inner term closer to the origin. Similarly, $\mu(0.25)$ will increase the update step of the positive samples. We show the results in Fig. 10.

$\alpha = 0.25$ leads to model divergence, as the negative samples have a stronger "pushing force" than the "pulling force" of positive samples in this setup. $\alpha = 0.75$ leads to lower discrepancy, as it increases the "pulling force" of positive samples, implicitly decreasing the effect of divergence penalty.

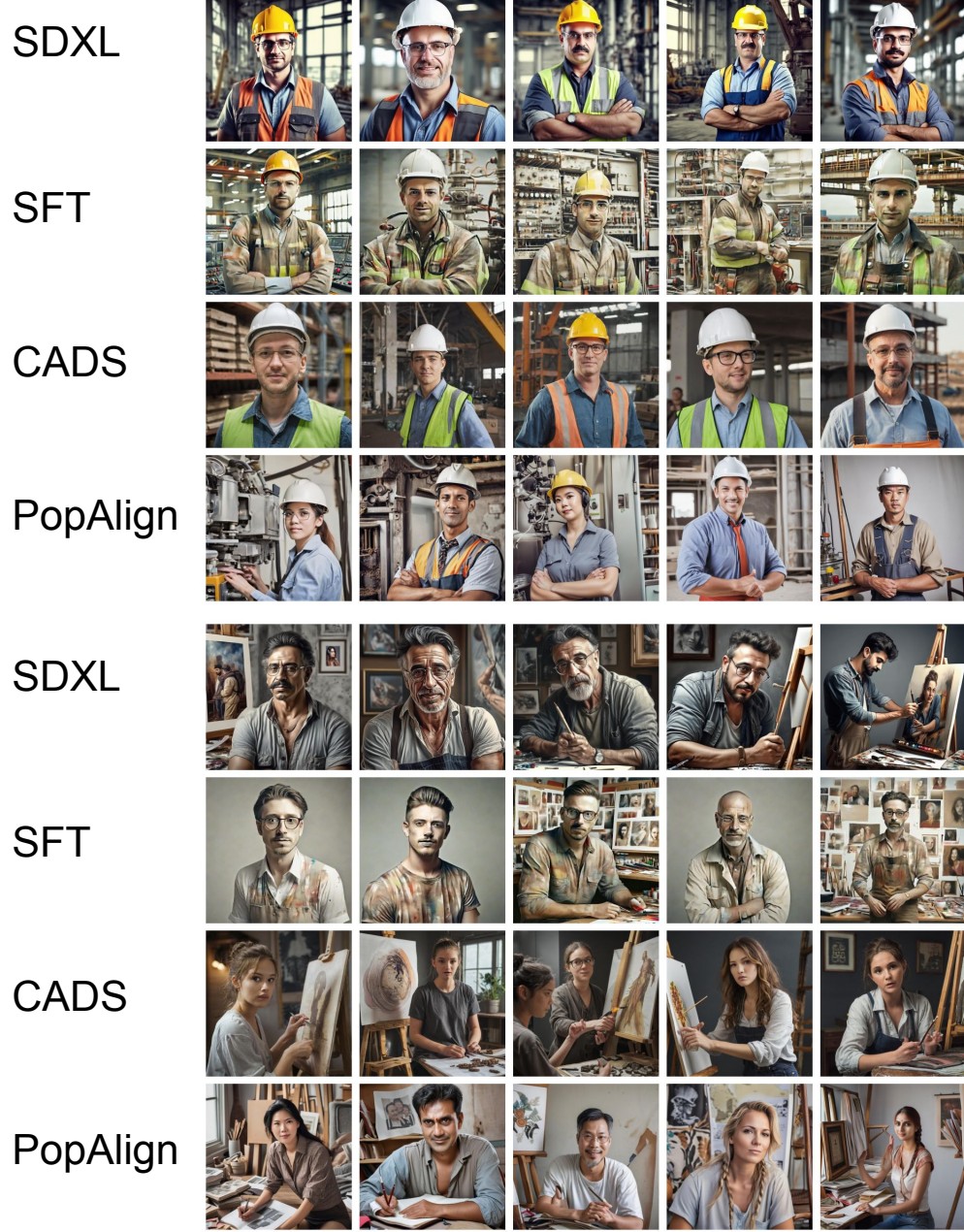

**Figure 8: Additional qualitative results on gender-neutral prompts.** PopAlign offers a diverse representation of identities while maintaining a comparable image quality with the SDXL baseline. The top four rows are generated using the prompt "engineer", while the bottom four rows are generated using the prompt "artist". The prompts are formatted in "best quality, a realistic photo of [prompt]"

## K  Licenses

We makes use the following models: CLIP (MIT license), PickScore(MIT license), HPS v2 (Apache-2.0 license), LAION Aesthetics predictor (MIT license), Deepface (MIT license), SDXL(CreativeML Open RAIL++-M License). Diffusion-DPO (Apache-2.0 license).

We use prompts from Pick-a-Pick dataset (MIT License).

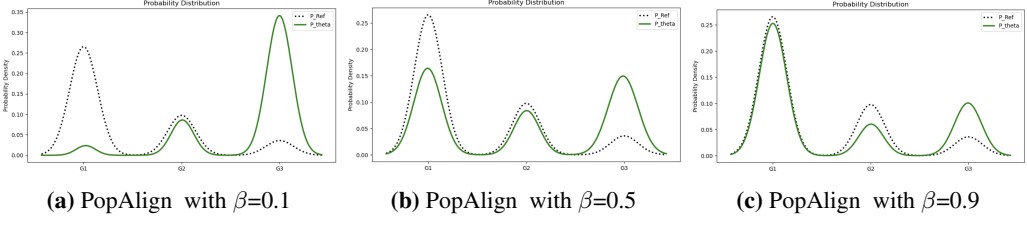

**(a)** PopAlign with $\beta$=0.1       **(b)** PopAlign with $\beta$=0.5       **(c)** PopAlign with $\beta$=0.9

**Figure 9:** Effect of $\beta$ in PopAlign

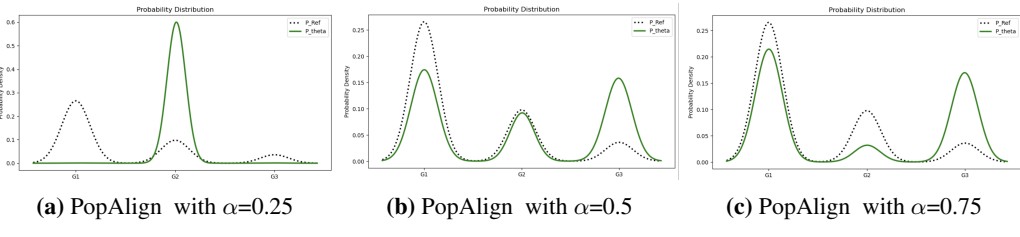

**(a)** PopAlign with $\alpha$=0.25       **(b)** PopAlign with $\alpha$=0.5       **(c)** PopAlign with $\alpha$=0.75

**Figure 10:** Effect of $\alpha$ in PopAlign

