# OpenReview forum: "PopAlign: Population-Level Alignment for Fair Text-to-Image Generation"
_NeurIPS.cc/2024/Workshop/SafeGenAi — SafeGenAi Poster_

### Official Review · Reviewer_oSwc · 2024-10-08
**PopAlign is a good paper for fair in T2I generation, with soild math. This paper makes contribution to developing a new DPO method for diffusion model alignment, meanwhile keeps their method computationally cheaper.**

**Rating:** 8
**Confidence:** 4

**Review:**

PopAlign introduces a Pop-level DPO method. It uses rigorous mathematical derivations to make this method computationally cheaper, and maintains high quality of image generation. With high quality, clarity and originality, PopAlign does make contributions to developing fair Text-to-Image generation models.
Weakness:
1.some problems maybe remain to be explored.For example, Can "Fairness-Alignment" be transferred?(e.g.,If you align the model to generate  nearly equal number of male and female doctors, does the model transfer this capability to generate  nearly equal number of male and female teachers?)
2.Some typos need to be addressed: You can see ?? in line 277,289

---

### Official Review · Reviewer_5ECY · 2024-10-09
**Promising study but need some clarification for experiment details**

**Rating:** 7
**Confidence:** 4

**Review:**

This paper presents a population-level preference optimization to solve the issue that sample level RLHF/DPO cannot fix some bias in text-to-image generation.

It is an insightful observation that the sample-level optimization works for LLM but not for T2I.

The experiments are well designed and show the improvement made by the new approach. While there are sufficient comparisons, some abbreviations are a bit vague - especially the relationship between “SDXL-SFT,” “SDXL-PopAlign,” “SDXL-DPO,” and “SDXL-DPO-PopAlign.”
My assumption is that:
SFT on top of SDXL -> SDXL-SFT
PopAlign  on top of SDXL -> SDXL-PopAlign
DPO  on top of SDXL -> SDXL-DPO
PopAlign  on top of SDXL-DPO -> SDXL-DPO-PopAlign
I suppose that SFT and DPO need different datasets - adding these details in the appendix will be helpful.
In addition, it is also worth comparing SDXL-DPO and SDXL-PopAlign, since PopAlign is inspired by DPO. Based on the table, DPO may still have advantage in individual quality and so PopAlign surpasses the baselines as an add-on but not a replacement.

The training data assumption is a bit unclear since Appendix B.1 does not share the prompt to generate prompts. The examples in the paper always contain only a single person and the prompt is related to profession.

For evaluation, the efforts of “Identity-Specific Prompts” are really good. Some bonus exploration may be to study some challenging cases, i.e. “Identity-Implicative Prompts.” For example, whether the discrepancy of race will be impacted by culture-related prompts.

Last, there are some “??” - possibly broken links (Line 277 and 289)

Overall: promising study but need some clarification for experiment details.